# Bioactive Polyphenols from Southern Chile Seaweed as Inhibitors of Enzymes for Starch Digestion

**DOI:** 10.3390/md18070353

**Published:** 2020-07-08

**Authors:** Luz Verónica Pacheco, Javier Parada, José Ricardo Pérez-Correa, María Salomé Mariotti-Celis, Fernanda Erpel, Angara Zambrano, Mauricio Palacios

**Affiliations:** 1Graduate School, Faculty of Agricultural Sciences, Universidad Austral de Chile, Valdivia 5090000, Chile; luz_pacheco007@hotmail.com; 2Institute of Food Science and Technology, Faculty of Agricultural Sciences, Universidad Austral de Chile, Valdivia 5090000, Chile; 3Department of Chemical and Bioprocess Engineering, Pontificia Universidad Católica de Chile, Macul, Santiago 7810000, Chile; perez@ing.puc.cl (J.R.P.-C.); faerpel@uc.cl (F.E.); 4Programa Institucional de Fomento a la Investigación, Desarrollo e Innovación, Universidad Tecnológica Metropolitana, Santiago 8940577, Chile; mmariotti@utem.cl; 5Instituto de Bioquímica y Microbiología, Facultad de Ciencias, Universidad Austral de Chile, Valdivia 5090000, Chile; angara.zambrano@uach.cl; 6Centro FONDAP de Investigación Dinámica de Ecosistemas Marinos de Altas Latitudes (IDEAL), Valdivia 5090000, Chile; mauricio.palacios@alumnos.uach.cl; 7Programa de Doctorado en Biología Marina, Facultad de Ciencias, Universidad Austral de Chile, Valdivia 5090000, Chile; 8Facultad de Ciencias, Universidad de Magallanes, Punta Arenas 6200000, Chile

**Keywords:** seaweed polyphenols, hypoglycemic effect, starch digestion, enzyme inhibition, cochayuyo

## Abstract

The increment of non-communicable chronic diseases is a constant concern worldwide, with type-2 diabetes mellitus being one of the most common illnesses. A mechanism to avoid diabetes-related hyperglycemia is to reduce food digestion/absorption by using anti-enzymatic (functional) ingredients. This research explored the potential of six common Chilean seaweeds to obtain anti-hyperglycemic polyphenol extracts, based on their capacity to inhibit key enzymes related with starch digestion. Ethanol/water hot pressurized liquid extraction (HPLE), which is an environmentally friendly method, was studied and compared to conventional extraction with acetone. Total polyphenols (TP), antioxidant activity, cytotoxicity and inhibition capacity on α-glucosidase and α-amylase were analyzed. Results showed that the *Durvillaea antarctica* (cochayuyo) acetone extract had the highest TP content (6.7 ± 0.7 mg gallic acid equivalents (GAE)/g dry seaweed), while its HPLE ethanol/water extract showed the highest antioxidant activity (680.1 ± 11.6 μmol E Trolox/g dry seaweed). No extract affected cell viability significantly. Only cochayuyo produced extracts having relevant anti-enzymatic capacity on both studied enzymes, showing a much stronger inhibition to α-glucosidase (even almost 100% at 1000 µg/mL) than to α-amylase. In conclusion, from the Chilean seaweeds considered in this study, cochayuyo is the most suitable for developing functional ingredients to moderate postprandial glycemic response (starchy foods), since it showed a clear enzymatic inhibition capacity and selectivity.

## 1. Introduction

Seaweeds or macroalgae are the most important benthic organisms in coastal marine ecosystems. According to their specific pigments, algae are classified into three divisions. The first group is the so-called brown algae or Ochrophyta (Phaeophyceae class), whose pigmentation varies from brown yellow to dark brown and produces a large amount of protective mucus. Red algae or Rhodophyta is the second largest group of algae and is found in various media. Finally, green algae or Chlorophyta is less common than brown and red algae. Its pigmentation varies from greenish yellow to dark green [1]. Besides its chemical composition, seaweeds have been studied as sources of a variety of compounds with potential biological activities such as antitumoral, antidiabetic and antioxidant, among others. These bioactive compounds are synthesized according to the level of maturity and capacity of the plant to interact with environmental conditions such as radiation, water pressure and salinity, which make them particularly attractive [2]. For example, phlorotannins (the polyphenols found in brown algae) comprise oligomers or polymers of phloroglucinol (1,3,5-trihydroxybenzene) with different antioxidant activities [3,4,5]. Moreover, it has been shown that these polyphenols from algae have significant in vitro inhibitory activities against α-glucosidase and α-amylase. This might have a potential application to control type-2 diabetes, since the inhibition of these enzymes would reduce the severity of postprandial hyperglycemia by delaying starch hydrolysis [6,7]. As an example, *Sargassum patens*, a brown alga of the Noto peninsula in the Ishikawa prefecture of Japan, contains 2-(4-(3,5-dihydroxyphenoxy)-3,5-dihydroxyphenoxy) benzene-1,3,5-triol DDBT, a phlorotannin with an inhibiting effect on the enzymes that hydrolyze carbohydrates (IC_50_ 3.2 µg/mL for α-amylase inhibition) [8]. The topography, waves, and exposure to wind of the benthic habitats of continental Chile favor the growth of seaweeds, where approximately 440 species have been identified [9]. Thus, Chilean seaweeds are an interesting source of new compounds with different applications.

Currently, the screening of the inhibitors of α-amylase and α-glucosidase of natural origin has received attention because this would avoid the side effects of commercial inhibitors for treating type-2 diabetes [10]. Extracts of natural plants with inhibitory α-glucosidase activity, such as tea and raspberry, are recommended as substitutes for synthetic drugs [11]. In addition to their anti-enzymatic activities, plant polyphenols are also capable of capturing free radicals and therefore act as antioxidants. It has been postulated that in diabetic patients, antioxidants help to prevent vascular diseases, the destruction of pancreatic cells and the formation of reactive oxygen species [12]. Seaweeds have recently gained significant interest as a sustainable source of various bioactive natural compounds, including polyphenols, carotene, lutein, astaxanthin, zeaxanthin, violaxanthin and fucoxanthin (pigments) [13]. Conventional processing technologies, based on organic solvent extraction at low pressures and temperatures, offer a simple approach for isolating such compounds; they have been used for a long time and are still widely applied. The main factors that should be considered to achieve an effective solid–liquid conventional extraction are the polarity of the compounds of interest, the characteristics of the solvent (toxicity, volatility, polarity, viscosity and purity) and the possible formation of other compounds during the extraction process. In addition, the yield, selectivity and contamination with undesired compounds are the usual performance indices used to compare different options. However, conventional techniques suffer from several limitations, such as being time-consuming and requiring large amounts of polluting or toxic solvents [14].

Alternative, faster and more efficient/selective techniques have been developed in the recent decades. Hot pressurized liquid extraction (HPLE), also known as accelerated solvent extraction, is a green alternative technique that has been widely applied to recover bioactive compounds from plant matrices. In this method, the extraction occurs at elevated temperatures and pressures (below the critical point of the solvent), normally in the ranges of 50–200 °C and 35–200 bar, respectively. Under these conditions, the viscosity and surface tension of the solvent are significantly reduced, whereas the solubility and mass transfer of the solute are greatly enhanced. Consequently, HPLE is fast, reduces the solvent consumption and allows for an efficient usage of green solvents such as water and ethanol for the extraction of a variety of compounds by changing their solvation dielectric constants (polarity) to values similar to those of organic solvents [15]. 

The objective of this research was to explore the potential of common seaweeds present in southern Chile to obtain anti-glycemic polyphenol rich extracts for functional food development. HPLE was applied and compared with an acetone conventional extraction method.

## 2. Results and Discussion

### 2.1. Cytotoxicity Assay 

The growing interest in plant-derived compounds for commercial human applications requires the assessment of their possible toxicity. Cytotoxicity studies in cell lines can be considered as a first step in the development of pharmaceutical, cosmetic or food products; toxicity levels (minimal or no toxicity) should be verified [16]. In this test, HT-29 cells, the human colon adenocarcinoma cell line expressing the characteristics of mature intestinal cells, such as enterocytes or mucus-producing cells [17] were used. As seen in Figure 1, the incubation with extracts at low concentrations (1 and 10 μg/mL) did not reveal significant changes in the MTT conversion rates associated with cell viability. *Nothogenia* sp. and *M. laminarioides* extracts negatively affected the cellular metabolism of HT-29 cells at 24 and 48 h, respectively, at the maximum concentration evaluated (1000 μg/mL); however, no extract decreased the cell viability in the same way as the positive control (DMSO) that reduced viability to 17.1 ± 1.6% and 18.4 ± 1.4% at 24 and 48 h of exposure, respectively. In general, no statistically significant differences were observed between the type of extract (ethanol and acetone) or between the incubation times (24 and 48 h). According to Galindo et al. [18], a cytotoxic effect can be considered when the viability is less than 75%. None of the seaweed extracts reduced the cellular viability to that level, so they may be considered to not be cytotoxic.

The diversity of compounds derived from algae can generate different actions; these can stimulate growth or apoptosis. *Nothogenia* sp. and *Pyropia* sp. extracts slightly decreased cell viability at higher concentrations, therefore, they could be analyzed from a chemotherapeutic perspective. Seaweed extracts could also be used as antitumoral agents. Using different cell types can show different responses towards a specific compound or plant extract. Some authors indicate that polyphenols are toxic to certain types of cancer cells that proliferate rapidly but are nontoxic to others [19,20].

### 2.2. Total Polyphenol Content and Antioxidant Capacities 

Both the species and extraction method significantly affect (*p* < 0.05) the total polyphenol content (TP) of the extracts (Table 1). The seaweed with the highest TP was *D. antarctica* (cochayuyo), whose acetone and ethanol extracts contained 7.4 ± 0.2 and 6.7 ± 0.7 mg gallic acid equivalents (GAE)/g dry seaweed, respectively, followed by *Pyropia* sp., whose acetone and ethanol extracts contained 6.2 ± 0.2 and 4.8 ± 0.3 mg GAE/g, respectively. In most cases, conventional extraction with acetone yielded higher values of TP (10%–20%) than HPLE with ethanol (*p* < 0.05). Acetone, being a dipolar compound, has an intermediate polarity and solubilizes solutes with a similar relative polarity. Wang et al. [21,22] found that acetone (70%) was the most suitable solvent to recover algae polyphenols. Nonetheless, acetone is a toxic solvent not permitted for human consumption. However, HPLE with ethanol (50%) allowed for seaweed extracts that are of food grade, which is mandatory for future applications such as nutraceuticals and functional food ingredients.

Two complementary methods were applied to determine the antioxidant activity of the extracts: the oxygen radical absorbance capacity (ORAC) test, which corresponds to an electron transfer model, and the DPPH assay that determines the transfer of hydrogen atoms. 

Statistically significant differences were observed among the species studied (*p* < 0.05), with brown seaweed extracts being the ones with the highest antioxidant activities in both extraction methods. The high antioxidant activity of brown species could be due mainly to their content of phlorotannins, whose phenolic ring can eliminate the reactive species of oxygen [23,24,25,26].

Ethanol *D. antarctica* extracts presented the highest antioxidant activity, both in the ORAC (680.1 ± 11.0 μmol ET/g dry seaweed) and DPPH (48.5 ± 4.2 μmol ET/g dry seaweed) assays (Table 1). Thus, HPLE with ethanol is a feasible food grade alternative to conventional extraction since it allows obtaining polyphenol extracts with higher antioxidant activity (*p* < 0.05). It is worth mentioning that the TP extraction yields differed only between 10–20%. Contrarily, for the antioxidant activity, the HPLE extracts presented two and 10 times higher DPPH and ORAC values, respectively, as compared to the extracts obtained using acetone. The advantage of using HPLE with ethanol is clear; the antioxidant capacity is associated with the potential benefic effect of polyphenols on human health, and above all because ethanol is a food grade solvent.

The major compounds contributing to the overall antioxidant activity in seaweeds are frequently phenolic compounds and polysaccharides; they may be found alone or associated with other components such as polyphenols, amino acid, protein, lipids, and sometimes polysaccharide conjugates [27] (Jacobsen et al., 2019). Since some seaweed polyphenol compounds may be conjugated with different types of sugars, the chemical structure of these conjugated compounds makes them more soluble in alcoholic solvents compared to acetone. This may explain the higher antioxidant activity of our extracts. The higher solubility of sugars can be associated with the dielectric constant of ethanol (24.55 at 25 °C), which is closer to water (78.54 at 25 °C) than acetone (20.7 at 25 °C) [28].

### 2.3. Anti-Enzymatic Activities

One of the strategies to manage type-2 diabetes is through the inhibition of enzymes such as α-amylase and α-glucosidase. The selective inhibition of these enzymes reduces the hydrolysis of carbohydrates and the rate of glucose uptake into the bloodstream [29]. The ability of the seaweed extracts to inhibit the activity of α-amylase and α-glucosidase at different concentrations was determined.

The α-amylase activity did not decrease significantly with any of the ethanol seaweed extracts within the concentration range studied (Figure 2). Acetone extracts of *D. antarctica* and *Gelidium* sp. inhibited the activity of this enzyme, although to a lesser extent than acarbose. The acetone extract of *D. antarctica* (2000 µg of dry extract/mL) decreased the α-amylase activity to 56.6 ± 2.0% and *Gelidium* sp. to 77.9 ± 2.1%. In contrast, acarbose (concentration ≥ 1000 μg/mL) decreased the enzyme activity to 37.5 ± 0.4%.

Like α-amylase inhibitors, α-glucosidase inhibitors can slow down the breakdown and absorption of dietary carbohydrates [30]. The acetone and ethanol extracts of *D. antarctica* and the acetone extract of *L. spicata* were the most effective inhibitors of α-glucosidase (Figure 3). The *D. antarctica* acetone extract almost completely inhibited the enzymatic activity (to 0.7 ± 0.3%), followed by the acetone extract of *L. spicata* (to 1.2 ± 0.3%) and by the ethanol extract of *D. antarctica* (to 3.1 ± 0.4%); all of them to a concentration of 1000 μg dry extract/mL. These extracts were much more effective than the commercial enzymatic inhibitor acarbose (1000 μg/mL), which reduced the enzymatic activity only to 40.4 ± 1.1%. The ethanol and acetone extracts of *Gelidium* sp., *M. laminarioides* and *Nothogenia* sp. did not show any significant inhibitory effect against α-glucosidase in the concentration range assessed (1, 10, 100, 1000 μg dry extract/mL). In general, the differences in the inhibition capacity among the extracts would be derived from their compositions, which in turn is the result of the seaweed species and extraction methods used.

The IC_50_ values for the α-glucosidase (Table 2) of acetone extracts of *D. antarctica* and *L. spicata* (the only two seaweed species that generated inhibition higher than 50%) were 466 and 479.21 μg/mL, respectively, and both were lower than the IC_50_ of acarbose (797.9 μg/mL). Hence, *D. antarctica* and *L. spicata* have an effective inhibitory effect on the activity of α-glucosidase. Brown algae were better inhibitors of α-glucosidase than red algae such as *Gelidium* sp., *M. laminarioides* and *Nothogenia* sp. Likewise, other authors found that the water extract of *Laminaria digitata*, a member of the Laminariaceae family and a brown seaweed, was effective in inhibiting α-glucosidase [11].

The diversity and abundance of some compounds may hamper the interactions with the enzyme at a molecular level; however, it is possible that the purification of extracts may enhance the enzymatic inhibitory activity. Therefore, our crude algae extracts were not as effective in inhibiting these enzymes as the pure polyphenols of vegetable origin (IC_50_ ~ 5 μg/mL) [14] or purified polyphenol extracts from pomegranate extracts (IC_50_ of 278 μg/mL) or black tea (IC_50_ of 64 μg/mL) [31].

Several reports have been published on the established enzyme inhibitors, such as acarbose, miglitol, voglibose and nojirimycin, and their favorable effects on blood glucose levels after food intake [32,33]. This fact is attributed to their capacity to diminish the assimilation of sugars, both monosaccharides and more complex carbohydrates. It seems appropriate to ingest the potent inhibitors of α-glucosidase and α-amylase to control the release of glucose from saccharides in the intestine. However, the complete inhibition of both enzymes could result in the poor absorption of nutrients. Inhibitors should be administered in doses that allow all carbohydrates to be digested. Otherwise, undigested carbohydrates will enter the colon and, as a result of bacterial fermentation, will lead to side effects such as flatulence. The most adequate inhibitors should delay the digestion and absorption of carbohydrates without overloading the colon with them [34]. It has been proposed that the administration of products presenting the moderate inhibition of α-amylase and strong inhibition of α-glucosidase is an effective therapeutic strategy that could decrease the availability of monosaccharides for absorption in the intestine [35,36,37]. Hence, the consumption of some of our seaweed extracts may be a preferred alternative for modulating the glycemic index of some food products.

The enzymatic inhibition capacity (α-amylase and α-glucosidase) of seaweed extracts followed the same trend as their antioxidant activities. The ethanol and acetone extracts of *D. antarctica*, the species with the highest average antioxidant activity (ORAC and DPPH), presented the highest α-glucosidase inhibition activity. Consequently, antioxidant activity indices can be used to optimize the design of the extraction process.

## 3. Materials and Methods

### 3.1. Chemicals and Cell Culture

All the chemicals and cell culture reagents were obtained from Sigma Chemical Co. (Saint Louis, MO, USA) unless stated otherwise. The cell lines were obtained from The Institute of Biochemistry and Microbiology UACh (Valdivia, Chile).

### 3.2. Seaweed Collection and Identification

Six species (4 red and 2 brown) were harvested in September 2017 from Corral port at “Region de los Rios” South of Chile. The red species collected were: *Gelidium* sp., *Mazzaella laminarioides* (Luga cuchara), *Nothogenia* sp. (Chascuo) and *Pyropia* sp. (Luche); the brown species were: *Durvillaea antarctica* (cochayuyo) and *Lessonia spicata* (huiro negro) (Figure 4). All of them were morphologically identified by taking into account taxonomical characteristics such as the shape of the thallus, the number of pyrenoids, the presence/absence of marginal teeth, and the thickness of the cross section. The six seaweeds were identified by Ph.D (c). M. Palacios (Algae Ecophysiology Laboratory of the Universidad Austral de Chile, Valdivia, Chile), an experienced researcher on Sub Antarctic macroalgae taxonomy.

### 3.3. Cytotoxicity Assays

To ensure the safety of the extracts as edible products, the cytotoxicity was assessed according to Lordan et al. [11]. The HT-29 cell line (ATTC number: HTB-3) from human colon carcinoma was maintained in Dulbecco’s modified Eagle’s medium (DMEM), containing 10% fetal bovine serum (FBS), 50 U/mL penicillin, 50 mg/mL streptomycin and 2 mM L-glutamine. The cells were cultured at 37 °C with 5% CO_2_ in a humidified incubator. The cells in exponential growth were used.

The seaweed extracts were prepared like enzymatic assays and the stock solution was diluted with the culture medium to give the desired concentrations. For each assay, the HT-29 cells were seeded and acclimated for 24 h before treatment in a DMEM culture medium without phenol red, for which 100 μL of HT-29 cells were added per well (50,000 cells/well). Subsequently, 2 mL of increasing concentrations of algae extract were added (between 1 and 1000 mg seaweed extract/mL), done in triplicate; a positive control was used, that of a high concentration of DMSO (16.7% as a final concentration). Each microplate was incubated at 37 °C with 5% CO_2_ for 24 and 48 h. The MTT (3(4,5-dimethyl-2-thiazoyl)-2,5-diphenyltetrazolic) from Sigma-Aldrich (cat # M2128; Saint Louis, MO, USA) was used to evaluate the effect of different concentrations of extracts on the cells. The MTT reagent (5 mg/mL) was added 4 h before the end of the treatment. Then, a lysis buffer (50% dimethylformamide and 20% SDS) was added, homogenized and finally incubated 10–15 min to subsequently record its absorbance at 545 nm. Cell survival was determined using the MTT (3(4,5-dimethyl-2-thiazoyl)-2,5-diphenyltetrazolic) assay, which measures the change in metabolic activity, proportional to the number of viable cells.

### 3.4. Polyphenol Extraction Methods 

The collected seaweed samples were quickly washed in cold water to remove sand and other particles; they were immediately frozen and stored in vacuum-packed bags at −80 °C prior to freeze drying. Two methods of extraction for each of the seaweeds were employed: HPLE with ethanol/water (50%) and conventional extraction (at atmospheric conditions: 20 °C-14.69 psi) with acetone/water (60%) at atmospheric pressure. For both extraction methods, freeze-dried seaweed material was mixed using a 1:32 *w/v* ratio with the respective extraction solvent, ethanol (50% *v/v*) and acetone (60%). HPLE were carried out in an accelerated solvent extractor (Thermo Scientific™ Dionex™ ASE™ 150, Waltham, MA, USA) at 120 °C and 1500 psi. Each sample was subjected to one cycle of extraction of 20 min and using 150 mL of washing solvent. After extraction, water–ethanol (50%) extracts were transferred to an amber PET bottle and stored at −20 °C. Using a conventional water/acetone (60%) extraction, each seaweed sample was extracted for 1 h on a thermoregulatory rotary shaker at 100 rpm at 30 °C. Then, the mixture was centrifuged for 5 min at 6000 rpm and 20 °C, and the supernatant was transferred to a 50 mL flask wrapped in aluminum. A volume of water was added to the remaining solid, stirred manually and centrifuged. The washed supernatant was transferred to the previous flask and adjusted to 50 mL with a 60% water–acetone solution. Finally, the extract was transferred to an amber PET bottle and stored at −20 °C.

### 3.5. Total Polyphenol Content

The total polyphenol content of the extracts was determined using the Folin–Ciocalteau (FC) method, with gallic acid as the standard. In brief, 0.5 mL of the sample or solvent blank was diluted in 3.75 mL of distilled water. Subsequently, 0.25 mL of the FC reagent was added and homogenized. Then, 0.5 mL of the sodium carbonate solution (10% *w/v*) was added, and it was homogenized for 1 h at room temperature. The absorbance of the reaction product was measured at 765 nm (UV spectrophotometer 1240, Shimadzu, Kyoto, Japan). The total polyphenol content was calculated as a mg of gallic acid equivalents (mg GAE) per gram of dry seaweed, using an absolute standard curve of calibration in the range of 0.01–0.1 mg GAE/mL (r^2^). Each extract was analyzed in duplicate.

### 3.6. Free Radical Scavenging Using DPPH Radical

The quantification of the anti-radical activity, 2,2-diphenyl-1-picrylhydracil (DPPH) of the extracts, was carried out by the method of Tierney et al. [38] with minor modifications. Prior to each batch of analysis, a working solution of DPPH (0.048 mg/mL) was prepared by diluting a stock of 0.238 mg/mL prepared in methanol. In addition, three serial dilutions of the samples were performed with the extraction solvent, at concentrations in the range of 0.025–15 mg/mL. For the analysis, 0.5 mL of DPPH solution was added to microtubes with 0.5 mL of the extract’s dilutions. After homogenizing, the tubes reacted for 30 min at room temperature. Absorbance was measured at 520 nm on a UV 1240 spectrophotometer (Shimadzu, Kyoto, Japan). Likewise, the serial dilutions of Trolox were measured as the reference standard, and from which a standard curve was determined. The results were expressed in μmol equivalent of Trolox (ET)/g dry seaweed.

### 3.7. Free Radical Scavenging by Oxygen Radical Absorbance Capacity (ORAC) Assay

The ORAC method was carried out based on the procedure described by Caor and Prio [39], with minor modifications. The reaction was carried out in a 75 mM phosphate buffer (pH 7.4), in a 96-well microplate. Then, 45 μL of the sample and 175 μL of fluorescein were deposited at 108 nm. This mixture was incubated for 30 min at 37 °C; after that time, 50 μL of the AAPH solution was added to 108 mM. The microplate was immediately placed in the dual-scan microplate spectrofluorometer (Gemini XPS, San Jose, CA, USA) for 60 min; fluorescence readings were recorded every 3 min. The microplate was automatically shaken before and after each reading. For the calibration curve, Trolox was used at 6, 12, 18 and 24 μM. All reactions were carried out in triplicate. The area under the curve (AUC) was calculated for each sample by integrating the relative fluorescence curve (r^2^ > 0.99). The net AUC of the sample was calculated by subtracting the AUC of the blank. The regression equation between the net AUC and Trolox concentration was determined, and the ORAC values were expressed as µmol Trolox equivalents/g of dry seaweed (ET/g) using the standard curve established previously.

### 3.8. Inhibition of α-Amylase Activity

The ability of each extract to inhibit α-amylase activity was measured using the method described by Nampoothiri et al. [40] and adapted by Lordan et al. [11]. For the preparation of the samples, each extract was dried by aeration (using an aeration pump). After solvent evaporation, each dry sample was re-suspended in DMSO and filtered, obtaining a stock solution of 10 mg of dry seaweed extract/mL. From this stock, the assay dilutions were made: 0.01–2 mg/mL phosphate buffer (pH 6.9). Then, 100 μL of each sample dilution and 1% starch solution in 20 mM sodium phosphate buffer (pH 6.9 with 6 mM sodium chloride) were incubated in microtubes at 25 °C for 10 min in a water bath. A volume of 100 μL of porcine pancreatic α-amylase (0.5 mg/mL) was added to each tube and the samples were incubated at 25 °C for another 10 min. Then, 200 μL of dinitrosalicylic acid reagent were added and the tubes incubated at 100 °C for 5 min in a water bath. Subsequently, 50 μL of each reaction mixture was transferred to wells of a 96-well microplate and diluted by adding 200 μL of water to each well and the absorbance was measured at 540 nm in a microplate reader. The enzymatic activity was determined as follows:
(1)$$Enzymatic\;activity\;\left(\%\right)=\frac{Absorbance\;of\;extract}{Absorbance\;of\;control}\cdot 100$$
where the control is the enzyme–substrate reaction in the absence of inhibitors. The effect of the pharmacological inhibitor, acarbose, was also determined.

### 3.9. Inhibitions of α-Glucosidase Activity

The ability of each extract to inhibit α-glucosidase activity was measured using the method described by Nampoothiri et al. [40] and adapted by Lordan et al. [11]. The samples were prepared with the same methodology as the α-amylase activity assay. The inhibitory effect of each extract was measured at concentrations from 0.1 to 1000 μg/mL in 100 mM sodium phosphate buffer (pH 6.9). A volume of 50 μL of the extract solution and 50 μL of the 5 mM p-nitrophenyl-α-d-glucopyranoside (PNPG) solution (in a phosphate buffer) was mixed in a 96-well microplate and incubated at 37 °C for 5 min. Then, a phosphate buffer (100 μL) containing 0.1 U/mL of μ-glucosidase (from *S. cerevisiae*) was added to each well. The absorbance at 405 nm was recorded for 15 min using a microplate reader at 37 °C. The effect of the commercial inhibitor on the α-glucoside activity was also determined, and the data were processed as in the previous assay.

The IC_50_ value was also calculated, representing the concentration of the extract that caused 50% enzyme inhibition, which was calculated by linear regression analysis. That value was determined only when the inhibition was higher than 50%.

### 3.10. Statistics

All the data points were mean values of at least two or three independent experiments. Where appropriate, the data were analyzed by a one-way analysis of variance (ANOVA) followed by Tukey’s Multiple Comparison test. The software employed for the statistical analysis was STATGRAPHICS Centurion XV.II (Old Tavern Rd, The Plains, VA, USA).

## 4. Conclusions

Extraction is the first step in studies regarding plant bioactive compounds and plays a significant role in the final outcome. Currently, there is growing demand for sustainable extraction methods of bioactive compounds from plant sources. HPLE is a green method that allows obtaining food-grade plant extracts that have specific healthy properties. Using this technology, we were able to produce extracts of *D. antarctica* and *L. spicata* that showed higher antioxidant capacities than the extracts from red and green algae. From the extracts that we obtained in this study, only three (of two species) strongly inhibited α-glucosidase and one appreciably inhibited α-amylase. From the six species under study, *D. antarctica* stood out as particularly useful for developing an antihyperglycemic agent given its enzyme inhibition profile: the strong inhibition of α-glucosidase and moderate inhibition of α-amylase. The seaweed considered in this study showed no cytotoxicity under the tested conditions.

## Figures and Tables

**Figure 1 marinedrugs-18-00353-f001:**
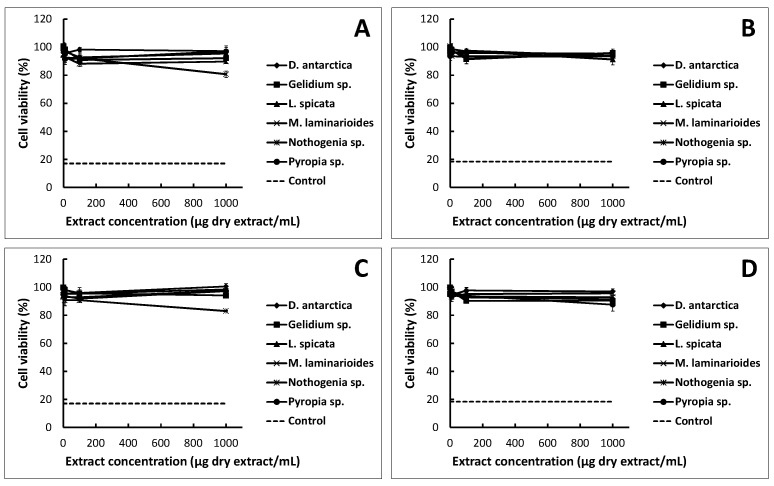
HT-29 cell viability at different dry extracts concentrations: (**A**) the ethanolic extract at 24 h incubation; (**B**) the ethanolic extract at 48 h incubation; (**C**) the acetone extract at 24 h incubation; (**D**) the acetone extract at 48 h incubation. Each point represents the mean of viable cells ± SD (*n* = 2). The dashed line is the cell viability using the positive control (DMSO 16.7%). The 100% viability was assigned to the cell culture without extracts.

**Figure 2 marinedrugs-18-00353-f002:**
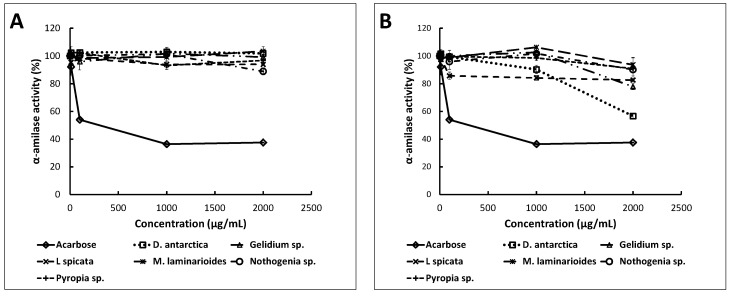
Percentages of the α-amylase activity under different concentrations of ethanol (**A**) and acetone (**B**) extracts (µg/mL). The points represent the average enzymatic activity (%) ± standard deviation (*n* = 3).

**Figure 3 marinedrugs-18-00353-f003:**
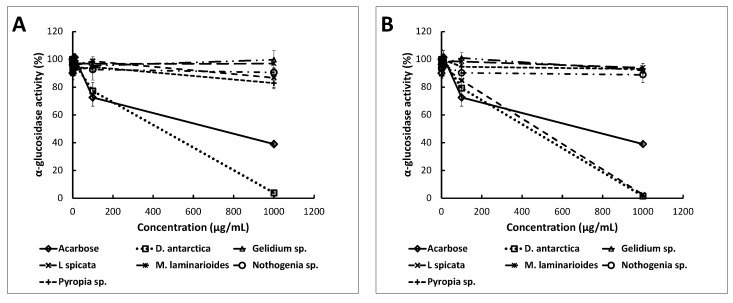
Percentages of α-glucosidase activity under different concentrations of ethanol (**A**) and acetone (**B**) extracts (µg/mL). Points represent the average enzymatic activity (%) ± standard deviation (*n* = 3).

**Figure 4 marinedrugs-18-00353-f004:**
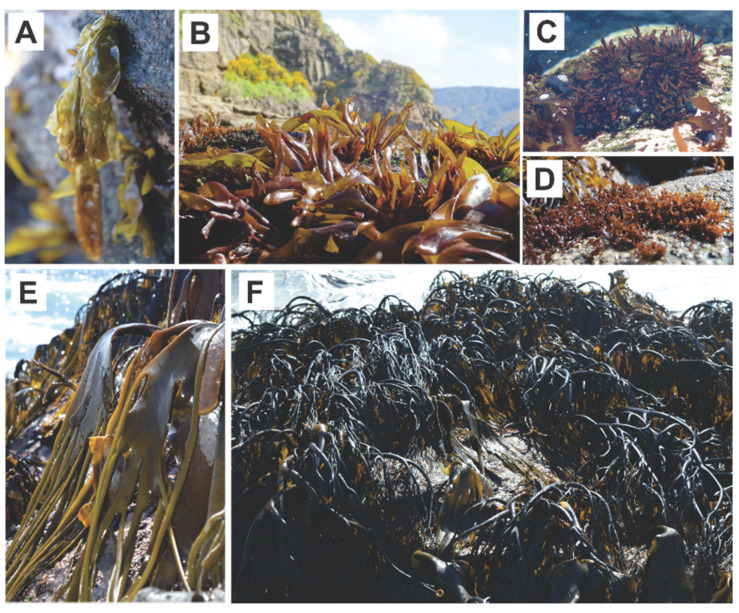
Macroalgae used in this study. The red macroalgae species (**A**) *Pyropia* sp., (**B**) *M. laminarioides*, (**C**) *Gelidium* sp., (**D**) *Nothogenia* sp.; and the brown macroalgae species (**E**) *D. antarctica* and (**F**) *L. spicata*. (Photograph by Mauricio Palacios- IDEAL Center).

**Table 1 marinedrugs-18-00353-t001:** Antioxidant activity of the seaweed extracts.

Species	Type	Total Polyphenolsmg GAE/g Dry Seaweed		DPPH μmol ET/g Dry Seaweed		ORAC μmol ET/g Dry Seaweed	
		Ethanol	Acetone	Ethanol	Acetone	Ethanol	Acetone
*Durvillaea antarctica*	Brown	7.4 ± 0.2 ^b^	6.7 ± 0.7 ^a^	48.5 ± 4.2 ^a^	27.8 ± 2.2^a^	680.1 ± 11.6 ^a^	64.7 ± 0.0 ^a^
*Gelidium* sp.	Red	3.2 ± 0.3 ^a^	3.4 ± 0.2 ^a^	4.8 ± 0.4 ^b^	4.7 ± 0.2 ^c^	277.8 ± 15.5 ^bc^	6.9 ± 0.5 ^c^
*Lessonia spicata*	Brown	3.3 ± 0.2 ^b^	3.8 ± 0.1 ^a^	6.6 ± 0.7 ^a^	10.7 ± 0.6 ^a^	448.3 ± 33.4 ^a^	21.3 ± 1.3 ^b^
*Nothogenia* sp.	Red	4.8 ± 0.3 ^b^	6.0 ± 0.3 ^a^	5.4 ± 0.3 ^ab^	6.9 ± 0.1 ^bc^	371.6 ± 12.3 ^ab^	18.1 ± 0.9 ^b^
*Mazzaella laminarioides*	Red	1.9 ± 0.1 ^a^	3.1 ± 0.1 ^a^	2.2 ± 0.1 ^c^	7.05 ± 0.8 ^b^	208.1 ± 10.4 ^c^	8.7 ± 0.6 ^c^
*Pyropia* sp.	Red	2.2 ± 0.0 ^b^	3.1 ± 0.1 ^a^	6.5 ± 0.4 ^ab^	10.6 ± 0.9 ^a^	455.3 ± 3.4 ^a^	30.7 ± 2.9 ^a^

Different letters indicate the statistically significant differences for the Tukey multiple range test with 95% confidence, into each column.

**Table 2 marinedrugs-18-00353-t002:** The IC50 values (μg extract/mL) of brown seaweeds for α-glucosidase inhibition.

Species	Extract Type	
	Ethanol	Acetone
*D. antarctica*	473.4 ± 0.9 ^b^	466 ± 1.3 ^a^
*L. spicate*	5317.6 ± 0.75 ^e^	479.2 ± 1.7 ^c^
Acarbose	797.85 ± 1.1 ^d^	

Different letters indicate the statistically significant differences for the Tukey multiple range test with 95% confidence.

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
