# Peer review of "Bioactive Polyphenols from Southern Chile Seaweed as Inhibitors of Enzymes for Starch Digestion"

_marinedrugs, 2020, doi:10.3390/md18070353_

Round 1
Reviewer 1 Report
The study “Bioactive polyphenols having anti-enzymatic activity related with starch digestion, extracted from southern Chile seaweed” represents an interesting report on the topic
However, I have several comments to suggest to the authors
Line 24: only reduce food digestion/absorption? Or also control with functional ingredients consumption
Line 192: why did the authors used HT29 cells ?
Line 231 Seaweed Collection and Identification: have the author’s detailed information on the characterization performed? Pictures/ numerical data?
Line 245: can the author provide detailed information on the preparation of seaweed stock solution? Are the stock solution diluted in acetone and or methanol? If so, did the author apply adequate viability controls to cell culture experiments?
Was the HPLE a good extraction emthod? This point should be better discussed in the revised version
Line 249: 1 and 1000 μg seaweed extract/mL: have these dosages physiological relevance? Why did the authors chose those conc? This point need to be discussed in the text.
Line 276-277: Why did the author s used Gallic acid as standard instead of Tannic acid equivalents?
Line 251: which MTT test did the authors performed? Which manufacturer?
Line 259: is the acetone extraction method the conventional method for seaweed? This point need to be discussed also in the discussion section as several extraction solvents have been previously used.
Fig 1: what is the 100% viability control? It should be specify in the legend
Table 1. the authors should uniform the units (eg mg trolox /g)
Author Response
Dear Reviewer, thank you very much for your comments. Below you can find responses for each one. Best regards.
Line 24: only reduce food digestion/absorption? Or also control with functional ingredients consumption
R: the sentence means that one of the mechanisms is to reduce food digestion/absorption by some factor, including the use of functional (anti-enzymatic) ingredients. For clarity sentence was changed.
Line 192: why did the authors used HT29 cells ?
R: Our principal idea was to test the cytotoxicity using intestinal cell lines, which is why we selected a cell line used in many studies about food additives or functional foods. As we mention in our work, we used HT-29 cell line from human colon adenocarcinoma cell line, which has received special interest in studies focused on food digestion and bioavailability due to the ability to express characteristics of mature intestinal cells; indeed, they can form a monolayer with tight junctions between cells and a typical apical brush border. They represent a valuable model due to their similarities with enterocytes of the small intestine; they have also been frequently used to study the intestinal immune response to bacterial infection, and microorganism survival, adhesion or invasion.
Line 231 Seaweed Collection and Identification: have the author’s detailed information on the characterization performed? Pictures/ numerical data?
R: identification was qualitatively done by an expert in the field. Manuscript was changed adequately, and one figure added as suggested.
Line 245: can the author provide detailed information on the preparation of seaweed stock solution? Are the stock solution diluted in acetone and or methanol? If so, did the author apply adequate viability controls to cell culture experiments?
R: As was written in the document stock solution was prepared like in enzymatic assay (section 3.8). For the preparation of the stock solution, each extract was dried by aeration (using an aeration pump). After solvent evaporation, each dried sample was re-suspended in DMSO and filtered obtaining a stock solution of 10 mg of dry seaweed extract /mL.
Was the HPLE a good extraction method? This point should be better discussed in the revised version
R: HPLE is a good method of extraction as was mentioned in the document. Although the extracts obtained using HPLE did not present the highest content of TP, they showed the highest antioxidant activity (680.1 ± 11.6 μmol E Trolox/g dry seaweed). In this sense it is worth mentioning that the differences between the TP extraction yields reached only between 10%-20%. Contrarily, for antioxidant activity HPLE extracts presented 2 and 10 times higher DPPH and ORAC values, respectively than the extracts obtained using acetone. Therefore, considering that antioxidant capacity is the property of polyphenols associated with their potential benefic effect on human health, the superiority of HPLE with ethanol is clear, moreover because the ethanol is a food grade solvent.
The discussion regarding the comparison between HPLE and acetone conventional method was included in the corrected version of the document.
Line 249: 1 and 1000 μg seaweed extract/mL: have these dosages physiological relevance? Why did the authors choose those conc? This point needs to be discussed in the text.
R: These dosages are not necessarily of physiological relevance, since in vivo analysis were not performed (that was not the focus of our research). The paper is focused on the in vitro effectiveness of the seaweed extracts that is why we choose these concentrations. Further studies to understand the actual physiological effect of bioactive compounds from seaweeds are needed.
Line 276-277: Why did the author s used Gallic acid as standard instead of Tannic acid equivalents?
R: Gallic acid equivalent is used when total polyphenols (all polyphenol types) are assessed. Considering that brown seaweeds not only contains phlorotannins, but large amounts of other type of polyphenols, Gallic acid is adequate.
Line 251: which MTT test did the authors performed? Which manufacturer?
R: We used MTT from Sigma- Aldrich (catalog number M2128), this test has been used by our group in several publications (doi: 10.3390/molecules23010014; doi: 10.1002/jcb.25790; among others). To complete this issue in the paper we included the specification of this reactive in materials and methods section.
Line 259: is the acetone extraction method the conventional method for seaweed? This point needs to be discussed also in the discussion section as several extraction solvents have been previously used.
R: When we talk about conventional extraction methods, it means at atmospheric conditions, both regarding temperature and pressure. Conventional extraction with acetone, is the most used polyphenol extraction method for different vegetal raw materials due to its solvent properties. As was mentioned in the document, the polarity of acetone is more affine to polyphenols than other solvents. Since we evaluated HPLE, as an alternative extraction method which is sustainable and used a food grade solvent, it was necessary to compare it with a yield extraction gold standard. Based on their solvent properties and previous results found in the literature on phlorotannin extraction, we decided to use acetone as a gold standard, although extraction optimization is not the focus of this paper.
We included this discussion in the corrected version of the document.
Fig 1: what is the 100% viability control? It should be specified in the legend
R: The 100 % viability means only cell culture without any type of extract. The legend is now completed in the revised version of this paper.
Table 1. the authors should uniform the units (eg mg trolox /g)
R: all corresponding units were changed to ET/g.
Reviewer 2 Report
Dear Authors,
The manuscript reporting the screening for potential α-glucosidase inhibiting crude extracts from seaweed samples collected at Southern Chile. The result showed a kelp, Durvillaea antarctica (cochayuyo) was the most active crude extract among others.
1) Since the specimen was identified by morphological features, I would suggest that Authors state the person's name that involved in this identification.
2) Can you explain why the Authors interested in checking the phenolic contents? Is there a relationship between α-glucosidase inhibiting activity of these samples and phenolic compounds, since Authors searched for α-glucosidase inhibitor.
3) What is the Author's justification for using a human cancer cell line, HT-29 as a toxicity evaluation for the samples? In my opinion, tested samples that do not show cytoxicity to human cancer cell lines but might have toxicity to healthy mammalian cells such as Vero cells.
4) Just a suggestion, Authors might considered to continue the study, at least be able to locate the potential compound from seaweeds that show α-glucosidase activity (as Author's claimed).
Author Response
Dear Reviewer, thank you very much for your comments. Below you can find reply to each one. Best regards.
1) Since the specimen was identified by morphological features, I would suggest that Authors state the person's name that involved in this identification.
R: Person’s name that involved in this identification was stated in the manuscript as suggested by reviewer.
2) Can you explain why the Authors interested in checking the phenolic contents? Is there a relationship between α-glucosidase inhibiting activity of these samples and phenolic compounds, since Authors searched for α-glucosidase inhibitor.
R: There is scientific evidence that many different polyphenols can regulate carbohydrate metabolism. For example Rasouli et al (2017) determined that compounds such: caffeic acid, curcumin, cyanidin, daidzein, epicatechin, eridyctiol, ferulic acid, hesperetin, narenginin, pinoresinol, quercetin, resveratrol and syringic acid can significantly inhibit the α-glucosidase enzyme and catechin, hesperetin, kaempferol, silibinin and pelargonidin are potent α-amylase inhibitors; all of them are polyphenols. That is why the phenolic content is relevant to understand at least in part the origin of any biological activity of the extracts.
3) What is the Author's justification for using a human cancer cell line, HT-29 as a toxicity evaluation for the samples? In my opinion, tested samples that do not show cytoxicity to human cancer cell lines but might have toxicity to healthy mammalian cells such as Vero cells.
R: Our principal idea was to test the cytotoxicity using intestinal cell lines, which is why we selected a cell line used in many studies about food additives or functional foods. As we mention in our work, we used HT-29 cell line from human colon adenocarcinoma cell line, which has received special interest in studies focused on food digestion and bioavailability due to the ability to express characteristics of mature intestinal cells; indeed, they can form a monolayer with tight junctions between cells and a typical apical brush border. They represent a valuable model due to their similarities with enterocytes of the small intestine; they have also been frequently used to study the intestinal immune response to bacterial infection, and microorganism survival, adhesion or invasion. We are agreed with the reviewer, indeed, to test the cytotoxicity in other cells lines is something that will have to do in the future.
4) Just a suggestion, Authors might considered to continue the study, at least be able to locate the potential compound from seaweeds that show α-glucosidase activity (as Author's claimed).
R: thank you very much for this suggestion, in fact we are obtaining purified extracts and performing MS/MS analysis to identify those compounds present in the most active extracts. We expect to have these results as soon as the laboratories reopen.
Round 2
Reviewer 2 Report
Dear Authors,
Yes, I can see that the phenolic content is relevant to understand at least in part the origin of biological activity within this work.
Thank you for a long justifying this work for using human cancer cell line, HT-29 as cytotoxicity evaluation.
I felt that the manuscript can be accepted in the present form.
Author Response
Thank you very much for comments.